# Selective Inhibition of Human Monoamine Oxidase B by Acacetin 7-Methyl Ether Isolated from *Turnera diffusa* (Damiana)

**DOI:** 10.3390/molecules24040810

**Published:** 2019-02-23

**Authors:** Narayan D. Chaurasiya, Jianping Zhao, Pankaj Pandey, Robert J. Doerksen, Ilias Muhammad, Babu L. Tekwani

**Affiliations:** 1National Center for Natural Products Research, Research Institute of Pharmaceutical Sciences, School of Pharmacy, The University of Mississippi, Oxford, MS 38677, USA; nchaurasiya@southernresearch.org (N.D.C.); jianping@olemiss.edu (J.Z.); rjd@olemiss.edu (R.J.D.); 2Department of BioMolecular Sciences, Division of Medicinal Chemistry and Research Institute of Pharmaceutical Sciences, School of Pharmacy, The University of Mississippi, Oxford, MS 38677, USA; ppandey@olemiss.edu

**Keywords:** acacetin 7-methyl ether, acacetin, monoamine oxidase-A, monoamine oxidase-B, molecular docking, molecular dynamics, neurological disorder, *Turnera diffusa*

## Abstract

The investigation of the constituents that were isolated from *Turnera diffusa* (damiana) for their inhibitory activities against recombinant human monoamine oxidases (MAO-A and MAO-B) in vitro identified acacetin 7-methyl ether as a potent selective inhibitor of MAO-B (IC_50_ = 198 nM). Acacetin 7-methyl ether (also known as 5-hydroxy-4′, 7-dimethoxyflavone) is a naturally occurring flavone that is present in many plants and vegetables. Acacetin 7-methyl ether was four-fold less potent as an inhibitor of MAO-B when compared to acacetin (IC_50_ = 50 nM). However, acacetin 7-methyl ether was >500-fold selective against MAO-B over MAO-A as compared to only two-fold selectivity shown by acacetin. Even though the IC_50_ for inhibition of MAO-B by acacetin 7-methyl ether was ~four-fold higher than that of the standard drug deprenyl (i.e., Selegiline^TM^ or Zelapar^TM^, a selective MAO-B inhibitor), acacetin 7-methyl ether’s selectivity for MAO-B over MAO-A inhibition was greater than that of deprenyl (>500- vs. 450-fold). The binding of acacetin 7-methyl ether to MAO-B was reversible and time-independent, as revealed by enzyme-inhibitor complex equilibrium dialysis assays. The investigation on the enzyme inhibition-kinetics analysis with varying concentrations of acacetin 7-methyl ether and the substrate (kynuramine) suggested a competitive mechanism of inhibition of MAO-B by acacetin 7-methyl ether with Ki value of 45 nM. The docking scores and binding-free energies of acacetin 7-methyl ether to the X-ray crystal structures of MAO-A and MAO-B confirmed the selectivity of binding of this molecule to MAO-B over MAO-A. In addition, molecular dynamics results also revealed that acacetin 7-methyl ether formed a stable and strong complex with MAO-B. The selective inhibition of MAO-B suggests further investigations on acacetin 7-methyl as a potential new drug lead for the treatment of neurodegenerative disorders, including Parkinson’s disease.

## 1. Introduction

The monoamine oxidases (EC.1.4.3.4; MAO-A and MAO-B) are FAD (flavin adenine dinucleotide)-dependent enzymes that are responsible for the metabolism of neurotransmitters, such as dopamine, serotonin, adrenaline, and noradrenaline, and for the oxidative deamination of exogenous arylalkyl amines [1,2]. Due to their central role in neurotransmitter metabolism, these enzymes represent attractive drug targets in the pharmacological therapy of neurodegenerative diseases and neurological disorders [3,4]. Recent efforts toward the development of MAO inhibitors have focused on selective MAO-A or MAO-B inhibitors. Selective MAO-A inhibitors are effective in the treatment of depression and anxiety [5], whereas the MAO-B inhibitors are useful for treatment Parkinson’s disease and in combination for treatment of Alzheimer’s Disease [4,6,7,8,9]. In recent studies, acacetin was reported as a potent inhibitor of MAO-A and MAO-B isolated from *Calea urticifolia* [10]. Similarly, acacetin and its derivative, acacetin 7-*O*-(6-*O*-malonylglucoside), from *Agatache rugosa* was also reported as a selective potent MAO-B reversible inhibitor [11].

*Turnera diffusa* Willd. ex Schult (known as damiana), which is a native plant to America and Africa, is traditionally used for the treatment of various diseases, including sexual impotence, neurasthenia, diabetes mellitus, urine retention, malaria, diarrhea, and peptic ulcer [12]. The plant is particularly used as a stimulant, an aphrodisiac, and generally as a tonic in neurasthenia and impotency with long tradition in Central America [13]. Phytochemical studies have revealed flavonoids, cyanogenic glycosides, terpenoids, and other secondary metabolites as prominent constituents in *Turnera diffusa* [14,15]. Several flavonoids from plant sources have been identified as inhibitors of MAO-A and MAO-B [16]. Based on the fact that damiana contains many flavonoids and that it is traditionally used as a tonic herb, we postulated that some components of the plant might be associated with the inhibition of MAOs. In the present study, the constituents that were isolated from damiana were evaluated for their inhibitory activities against recombinant human MAO-A and MAO-B. Acacetin 7-methyl ether showed the most potent selective inhibition of the MAO-B enzyme. The studies were further extended to investigate the selective binding and mode of interaction of acacetin 7-methyl ether with human MAO-B.

## 2. Results

### 2.1. Determination of MAO-A and -B Inhibition Activity

A series of flavonoids and flavonoid glycosides, namely acacetin, acacetin 7-methyl ether, vetulin (Figure 1), apigenin-7-*O*-β-d-(6-*O*-pcoumaroyl) glucoside, echinaticin, tetraphyllin B, tricin-7-glucoside, diffusavone, turneradiffusin, rhamnosylorientin, rhamnosylvitexin, and turneradin were isolated from *T. diffusa* [12].

Besides acacetin, acacetin 7-methyl ether and vetulin also showed selective concentration-dependent inhibition of MAO-B (Table 1, Figure 2). The MAO inhibitory properties of acacetin have been recently reported [10]. Acacetin 7-methyl ether was >500-fold selective for MAO-B (IC_50_ = 0.198 μM) as compared to MAO-A (IC_50_ = >100 μM) (Table 1). The concentration-dependent inhibition of MAO-B with acacetin 7-methyl ether showed a plateau at ~80% inhibition (~20% activity remaining). This may potentially be due to the low solubility of acacetin 7-methyl ether in assay buffer medium at higher concentrations. Even though the potency of inhibition of human MAO-B by acacetin 7-methyl ether was about four-fold lower when compared to the standard drug deprenyl (a selective MAO-B inhibitor), its selectivity for MAO-B was higher (>500 fold) as compared to deprenyl (450 fold). Deprenyl (available with the trade name selegiline) is a clinically used drug for the treatment of Parkinson’s disease and major depressive disorder [17]. Other constituents that were isolated from *T. diffusa* only showed moderate inhibition of MAO-A and MAO-B (IC_50_ in the range of 13–61 μM), with no significant selectivity towards MAO-B or MAO-A (Table 1). The studies were extended to investigate the kinetics of inhibition of MAO-B by acacetin 7-methyl ether. Further, characteristics of the interaction and the putative binding mode of acacetin 7-methyl ether were also investigated using computational docking of the ligand to the X-ray crystal structures of MAO-A and MAO-B.

### 2.2. Enzyme Kinetics and Mechanism Studies

Acacetin 7-methyl ether was tested against MAO-B at varying concentrations of kynuramine, which is a nonselective substrate, to investigate the nature of inhibition of the enzyme. Based on dose-response inhibition, five concentrations of acacetin 7-methyl ether were selected, two below the IC_50_ value (100 and 150 nM), one around the IC_50_ value (200 nM), and two above the IC_50_ value (300 nM and 500 nM) for the inhibition kinetics experiments. The assays were run at varying concentrations of the substrate and fixed concentrations of the inhibitor. The Ki (i.e., inhibition/binding affinity) values were computed with double reciprocal Lineweaver–Burk plots. The binding of acacetin 7-methyl ether to human MAO-B affected the K_m_ value (i.e., the affinity of the substrate for the enzyme) without much effects on the V_max_ (maximum enzyme activity), indicating that the inhibition of MAO-B by acacetin 7-methyl ether was competitive (Figure 3). Acacetin 7-methyl ether inhibited the enzymatic activity of MAO-B with considerably high affinity (Ki = 45 nM) (Table 2).

### 2.3. Analysis of Reversibility of Binding of Inhibitor

The characteristics of binding of acacetin 7-methyl ether to MAO-B were also investigated using an equilibrium-dialysis assay. The inhibitor acacetin 7-methyl ether was incubated at1.0 and 2.0 μM) with the MAO-B enzyme for 20 min and the resulting enzyme-inhibitor complex preparation was dialyzed overnight in phosphate buffer. The activity of the enzyme was analyzed before and after the dialysis (Figure 4). The incubation of MAO-B with 1.0 and 2.0 μM of acacetin 7-methyl ether caused >60% inhibition of activity and only ~80% activity of the enzyme was recovered after equilibrium dialysis. Thus, the binding of acacetin 7-methyl ether to MAO-B was partially reversible. The binding of selective MAO-B inhibitor deprenyl was confirmed to be irreversible (Figure 4).

### 2.4. Time-Dependent Assay for Enzyme Inhibition

The time dependence of binding of acacetin 7-methyl ether to MAO-B was analyzed. The recombinant enzyme was incubated for different times with the test compounds, namely, deprenyl, (0.100 μM), acacetin 7-methyl ether (0.500 μM), and acacetin (0.100 μM) (Figure 5). The control without inhibitors was also run simultaneously. The activity of the enzyme was determined, as described above, and the percentage of enzyme activity remaining was plotted against the pre-incubation time to determine the time-dependence of inhibition. The binding/inhibition of MAO-B by acacetin 7-methyl ether was not dependent on the pre-incubation time (Figure 5).

### 2.5. Computational Docking Study of Acacetin 7-Methyl Ether

Selective interactions of acacetin 7-methyl ether were investigated by molecular docking to understand its binding pose to *h*MAO-A and *h*MAO-B. Three-dimensional (3D) protein-ligand interactions of acacetin (Figure 6A,C, magenta, stick model) and acacetin 7-methyl ether (Figure 6B,D, cyan, stick model), respectively, with the X-ray crystal structures of MAO-A and MAO-B are presented in Figure 6. The docking scores and binding free-energies of acacetin 7-methyl-ether and acacetin at the active sites of the *h*MAO-A and *h*MAO-B X-ray crystal structures are shown in Table 3 and their corresponding putative binding poses are shown in Figure 6. The docking protocol was validated by self-docking, in which the native ligands, harmine, and pioglitazone, from the X-ray structures were docked into their corresponding protein’s structures of MAO-A and MAO-B, respectively. The calculated RMSD between the docked and experimental poses were <0.6 Å, which verified that the docking protocol was appropriate for this use.

According to the docking and binding free-energy results, acacetin 7-methyl ether showed better binding affinity (Docking score = −10.708 kcal/mol, ΔG = −67.494 kcal/mol) to *h*MAO-B than *h*MAO-A (Docking score = −9.085 kcal/mol, ΔG = −31.791 kcal/mol). The experimental data also supported this (Table 3). The docking results revealed that the binding orientation of acacetin 7-methyl ether and acacetin were similar to the orientations of the native ligands of the X-ray crystal structures of MAO-A (PDB ID: 2Z5X) and MAO-B (PDB ID: 4A79). The computational docking studies provided further insights into selective interactions of acacetin-7-methyl ether with the human MAO-B over MAO-A. Acacetin 7-methyl ether tightly binds to MAO-B by forming π-π stacking and H-bonding (at its C4 carbonyl) with Tyr326 and strong hydrophobic interactions with nearby amino acid residues. Ile199 and Ile316, which are known as critical residues for MAO-B selectivity [19], showed strong hydrophobic interaction with the acacetin 7-methyl ether. The additional methoxy moiety at the C7 position of acacetin 7-methyl ether compared to acacetin participated in the hydrophobic interactions with the Ile198. However, for MAO-A, this additional methoxy moiety exhibited bad contact with Phe352 and Lys305 and this may be the reason for the poorer binding with MAO-A and the selectivity of acacetin-7-methyl ether for MAO-B over MAO-A. Additionally, the conserved water molecules within the enzyme active site were important for enhancing the interaction of acacetin-7-methyl ether with MAO-B. Water-mediated hydrogen bonding was found between acacetin-7-methyl ether and Tyr188, Tyr435, and Gln65, which was similar to what was found for acacetin binding with MAO-B (Figure 6C). We also observed water-mediated H-bonding with the carbonyl of Cys172 and the oxygen at the C7 position of acacetin and acacetin-7-methyl ether (the interaction is not shown in Figure 6). A previous study on the interaction of acacetin with Cys172 reported direct (not water-mediated) interaction with Cys172 [11]. Interestingly, in the docked pose of acacetin with MAO-B that was reported in that study, the chromene moiety was flipped, with its carbonyl pointed towards Cys172. By contrast, in the docked pose that was determined in the present study, the carbonyl was pointed away from Cys172. To help determine which docked pose is more likely correct, we studied the available structures in the Protein Data Bank and found the X-ray crystal structure of MAO-B with co-crystalized ligand dimethylphenyl-chromone-carboxamide (PDB ID: 6FVZ) to be helpful. In 6FVZ, two alternate rotamers of Cys172 (side chain dihedral angle chi1 = −158° or −86°), each with an average occupancy of 50%, were included in the reported structure. The different values for the side chain dihedral angle of Cys172 open up the possibility of a direct H-bond interaction between Cys172 and acacetin or acacetin-7-methyl ether. The docked pose of acacetin that was reported in this study adopted a similar orientation for its chromene moiety (with the chromene oxygen being located close to Cys172) to that of the chromene moiety of the co-crystallized ligand of 6FVZ. These properties support the reliability of the docked pose that is presented here.

### 2.6. Molecular Dynamics Study of Acacetin 7-Methyl Ether and MAO-B

Docking alone cannot provide full insight into the binding mode and dynamics of acacetin 7-methyl ether within *h*MAO-B. Therefore, we carried out a 10 ns MD simulation of the complex of acacetin 7-methyl ether with MAO-B using Desmond software [20]. The MD simulation suggests that acacetin-7-methyl ether fits tightly into the binding pocket of MAO-B, since there was very little deviation in the ligand Root Mean Square Deviation (RMSD) (Figure 7) during the simulation. No significant flexibility (as measured by Root Mean Square (RMS) fluctuation, RMSF) was observed in the secondary structure elements (SSE) (α-helices and β-strand: Helix = 28.63%, Strand = 15.78%, and Total SSE = 44.40%) of the protein model, with major fluctuations only in residues 490–520 and the loop regions (Figure 8). The %SSE remained close to 45% throughout the simulation. The amino acids that interacted with acacetin-7-methyl ether did not show any major fluctuations in their RMSF values (Figure 8). The interaction histograms (Figure 9A) and protein-ligand contact graphs (Figure 9B) were analyzed to study the time-dependent changes in the interaction of the ligands with key residues of the protein. The interaction histogram shows that crucial amino acids for interaction with acacetin-7-methyl ether are Leu171 (hydrophobic and water bridges), Cys172 (hydrogen bond, hydrophobic, and water bridges), Tyr188 (hydrophobic and water bridges), Ile199 (hydrophobic), Gln206 (hydrogen bond and water bridges), Tyr326 (hydrophobic and water bridges), Tyr398 (water bridges), and Tyr435 (hydrogen bond, hydrophobic, and water bridges) (Figure 9A). The stabilization of the complex can be mainly attributed to several H-bond contacts and π-π interactions. H-bond contacts were observed between oxygen at the C7 position of acacetin-7-methyl ether and Cys172 and between the C5 OH and Tyr435.

Strong π-π interactions were observed between Tyr326 and Ring B and Ring C of acacetin 7-methyl ether. In addition, the protein-ligand interaction diagram (Figure 9B) indicates that acacetin 7-methyl ether forms H-bond interactions with Cys172 (70% contribution), Gln206 (24% contribution), and Tyr435 (30% contribution), and water-mediated hydrogen bond interactions with Leu171 (32% contribution) throughout the 10 ns MD simulations. The contribution of π-π interactions with Tyr326 was found to be 72% (with Ring C) and 24% (with Ring B). In conclusion, the interaction histograms and contact graphs show that acacetin 7-methyl ether forms a significantly stable complex with MAO-B.

## 3. Discussion

Acacetin 7-methyl ether (Figure 1) was identified as a potent and highly selective MAO-B inhibitory constituent of *Turnera diffusa*. Acacetin 7-methyl ether has also been isolated from a few other plants and showed antidiabetic, antiproliferative, and antioxidative activities [21,22,23]. MAO inhibitors, as well as oxidative stress reducers, have significantly contributed in the treatment of neurodegenerative disorders, including Parkinson’s disease [24]. However, MAO inhibitory action of acacetin 7-methyl ether has not been reported earlier.

The selective MAO-B inhibitors are used alone as well as in combination with carbidopa-levodopa for the treatment of motor symptoms of Parkinson’s disease [25]. Reduction in dopamine levels is responsible for the motor symptoms of Parkinson’s disease. The neurons that produce dopamine are damaged during the progression of Parkinson’s disease. Treatment with levodopa provides the precursor for the biosynthesis of dopamine. The addition of a MAO-B inhibitor to levodopa therapy results in a slowing of the breakdown of levodopa and dopamine in the brain, and it may boost the effect of levodopa. The current battery of clinically used MAO-B inhibitors includes selegiline (l-deprenyl), rasagiline, and safinamide. Selegiline and rasagiline are irreversible inhibitors of MAO and they may be associated with significant side effects [26,27]. Both acacetin, reported earlier, [10] and acacetin 7-methyl ether reported here show the potent inhibition of MAO-B. However, acacetin 7-methyl ether showed >500-fold selectivity, compared to only two-fold selectivity of acacetin for MAO-B over MAO-A. The 7-methyl substitution marginally reduces the potency of MAO-B inhibition, relative to acacetin (IC_50_ 198 nM vs. 50 nM, respectively), but it dramatically improves the selectivity (>500) for MAO-B. This suggests a significant scope for further optimization of the 5, 7-dihydroxy-4′-methoxyflavone pharmacophore to achieve high potency and selectivity against the human MAO-B. 5, 7-Dihydroxy-4′-methoxyflavones represent a new class of selective reversible MAO-B inhibitors with potential therapeutic application for treatment of neurodegenerative disorders, including Parkinson’s disease in combination with standard drugs [28]. A recent study has also reported MAO-A and MAO-B inhibition activity by acacetin 7-*O*-(6-*O*-malonylglucoside), a derivative of acacetin that was isolated from *Agastache rugosa* plant leaves [11]. The reversibility of MAO-B inhibition with acacetin 7-methyl ether adds significant value to its potential therapeutic use. A recent pharmacokinetics study of acacetin metabolism in vitro and in vivo that analysed its metabolites showed that acacetin is metabolized by the sulfotransferase 1A1 enzyme into acacetin-7-sulfate, which was detected in vivo in plasma samples and also in vitro from incubation of the Liver S9 fraction [29].

Further, computational analysis of the binding of acacetin 7-methyl ether to human MAO-A and MAO-B confirmed its selective and stronger interaction with MAO-B when compared to MAO-A. Both acacetin and acacetin 7-methyl ether interact with Gln65, Cys172, Tyr188, and Tyr435 in human MAO-B. However, less prominent interactions of acacetin 7-methyl ether with Tyr197 and Tyr407 in MAO-A as compared to acacetin afforded the higher inhibition of MAO-A with acacetin (IC_50_ = 0.115 μM) than with acacetin 7-methyl ether (IC_50_ = >100 μM). To shed light on the interaction profile, binding mechanism, and dynamic properties of acacetin 7-methyl ether with MAO-B, we carried out a 10 ns MD simulation of their protein-ligand complex. The structural and dynamical properties of acacetin 7-methyl ether were analyzed using RMSD plots of the ligand and of the Cα atoms of the protein (Figure 7), and by RMSF (Figure 8) plots of protein that was bound to the ligand. The RMSD plot of atom locations vs. simulation time indicated that the protein and ligand maintained a significantly stable state throughout the simulation (Figure 7). The most significant deviation in RMSD was a result of the flexible motion of the modeled C-terminal residues (residues 498–520) (not included in Figure 7). The MD simulations revealed that Leu171, Cys172, Tyr188, Ile199, Gln206, Tyr326, Tyr398, and Tyr435 are crucial amino acids for the interaction of acacetin 7-methyl ether and MAO-B. These results show us that acacetin 7-methyl ether forms a stable and strong complex with MAO-B.

These studies support the further in vivo evaluation of acacetin, when considering the MAO inhibitory activities that were reported earlier [10] and those of acacetin 7-methyl ether reported herein, in experimental animal models for neurological and neurodegenerative diseases.

## 4. Materials and Methods

### 4.1. Enzymes and Chemicals

Recombinant human monoamine oxidase-A and monoamine oxidase-B enzymes were obtained from the BD Biosciences (Bedford, MA, USA). Kynuramine, clorgyline, deprenyl, acacetin, and DMSO were purchased from Millipore Sigma (St. Louis, MO, USA). Safinamide was purchased from TCI Chemicals, Portland, OR, USA.

Acacetin 7-methyl ether (purity >98%), vetulin (>97%), apigenin-7-*O*-β-d-(6-*O*-pcoumaroyl) glucoside (>97%), echinaticin (>96%), tetraphyllin B (>98%), tricin-7-glucoside (>98%), diffusavone (>97%), turneradiffusin (>96%), rhamnosylorientin (>97%), rhamnosylvitexin (>95%), turneradin (>95%), and acacetin (>96%) were isolated from *T. diffusa* at the National Center for Natural Products Research (NCNPR), University of Mississippi, University, MS, USA. Their identities and purities were confirmed by chromatographic (TLC, HPLC) and spectroscopic (IR, one-dimensional (1D)- and two-dimensional (2D)-NMR, HR-ESI-MS) methods, as well as by comparison with the published spectroscopic data [14,15].

### 4.2. MAOs Inhibition Assay

To investigate the effect of the isolated constituents from *T. diffusa* on recombinant human MAO-A and MAO-B, the kynuramine oxidative deamination assay was performed in 384-well plates, as previously reported, with minor modifications [30,31]. A single fixed concentration of kynuramine substrate and varying concentrations of inhibitor were used to test enzyme inhibition and determine the IC_50_ values. Kynuramine concentrations for MAO-A and MAO-B were 80 μM and 50 μM, respectively. These concentrations of kynuramine were twice the apparent K_M_ value for substrate binding [10,31]. The concentrations that were tested for pure constituents were from 0.01 to 100 μM, for MAO-A and MAO-B inhibition assays. Reactions were performed in a clear 384-well microplate (50 μL) with 0.1 M potassium phosphate buffer, pH 7.4. The inhibitors and compounds were dissolved in DMSO, diluted in the buffer solution, and pre-incubated at 37 °C for 10 min (no more than 1.0% DMSO final concentration). The reactions were initiated by the addition of 12.50 μL MAO-A (to 5 μg/mL) or MAO-B (to 12.5 μg/mL), incubated for 20 min at 37 °C, and then terminated by the addition of 18.8 μL of 2N NaOH. The enzyme product 4-hydroxyquinoline formation was recorded fluorometrically using a SpectraMax M5 fluorescence plate reader (Molecular Devices, Sunnyvale, CA, USA) with an excitation (320 nm) and emission (380 nm) wavelength, using the SoftMaxPro 6.0 program. The inhibition effects of enzyme activity were calculated as a percent of product formation when compared to the corresponding control (enzyme-substrate reaction) without inhibitors. Controls, including samples in which the enzyme or the substrate was added after stopping the reactions, were simultaneously checked to determine the interference of inherent fluorescence of the test compounds with the measurements. IC_50_ values for MAO-A and MAO-B inhibition were calculated from the concentration dependent inhibition curves using XLfit, a Microsoft Excel-based plug-in which performs Regression, curve fitting and statistical analysis (IDBS, Bridgewater, NJ, USA).

### 4.3. Determination of IC_50_ Values

To determine the IC_50_ values for the inhibition of MAO-A and MAO-B by acacetin 7-methyl ether, the enzyme assay was performed at a fixed concentration of the substrate kynuramine for MAO-A (80 μM) and for MAO-B (50 mM) and with varying concentrations of inhibitor/test compounds for MAO-A and MAO-B (0.01 μM to 100 μM). The dose-response curves were generated using Microsoft^®^ Excel and the IC_50_ values were analyzed using XLfit software.

### 4.4. Enzyme Kinetics and Mechanism Studies

For the determination of the binding affinity of the inhibitor (Ki) with MAO-A and MAO-B, the enzyme assays were carried out at different concentrations of kynuramine substrate (1.90 μM to 500 μM) and varying concentrations of the inhibitors/compound. Compounds acacetin and acacetin 7-methyl ether were tested at 0.030–0.100 μM and 0.100–0.500 μM, respectively, for MAO-B to determine the Km and Vmax values of the enzymes in the presence of the inhibitor. The controls without inhibitor were also simultaneously run. The results were analyzed by standard double reciprocal Line–Weaver Burk plots for computing Km and Vmax values, which were further analyzed to determine Ki values [10].

### 4.5. Analysis of Reversibility of Binding of Inhibitor

The inhibitors bind with the target enzyme through the formation of an enzyme-inhibitor complex. The formation of the enzyme-inhibitor complex may be accelerated in the presence of the high concentration of the inhibitor. The reversibility of binding of MAO inhibitory compound acacetin 7-methyl ether was determined by the formation of the complex by incubating the enzyme with a high concentration of the inhibitor, followed by extensive dialysis of the enzyme-inhibitor complex and the recovery of catalytic activity of the enzymes. The MAO-B (0.2 mg/mL protein) enzyme was incubated with each inhibitor: acacetin (0.5 μM), acacetin 7-methyl ether (1.0, 2.0 μM), deprenyl (0.5 μM), and safinamide (0.2 μM) in a total volume of 1 mL, 100 mM potassium phosphate buffer (pH 7.4). After 20 min incubation at 37 °C, the reaction was stopped by chilling in an ice bath. All of the samples were dialyzed against potassium phosphate buffer (25 mM; pH 7.4) at 4 °C for 14–18 h (three buffer changes). Control enzyme (without inhibitor) was also run through the same procedure and the activity of the enzyme was determined before and after the dialysis [18].

### 4.6. Time-Dependent Assay for Enzyme Inhibition

To analyze whether the binding of the inhibitor with MAO-B was time-dependent, the enzyme was pre-incubated for different time periods (0–15 min) with the inhibitors at a concentration that produces approximately 70–80% inhibition. The inhibitor concentrations used to test time-dependent inhibition were acacetin (0.1 μM), acacetin 7-methyl ether (0.5 μM), and deprenyl (0.1 μM) with MAO-B (12.5 μg/mL). The controls without inhibitors were also simultaneously run. The activities of the enzymes were determined, as described above, and the percentage of enzyme activity remaining was plotted against the pre-incubation time to determine the time-dependence of inhibition.

### 4.7. Computational Methods

The X-ray crystal structures of MAO-A (PDB accession number: 2Z5X [32]) and MAO-B (PDB accession number: 4A79 [33]) were downloaded from the Protein Data Bank website. These proteins were prepared by adding hydrogens, adjusting bond orders and proper ionization states, and refining overlapping atoms. The water molecules beyond 5 Å from the co-crystalized ligands were removed and the ligand states were generated using Epik at pH 7.4. During the refinement process, the water molecules with less than two H-bonds to non-waters were also removed. Acacetin and acacetin-7-methyl ether were sketched in Maestro [34], prepared, and energy-minimized at a physiological pH of 7.4 using the LigPrep [35] module of the Schrӧdinger software (Cambridge, MA, USA) [36]. Acacetin was used as a positive control for the docking studies. For protein and ligand preparation, we used the Optimized Potentials for Liquid Simulations 3 (OPLS3) force field. The active sites of the MAO-A and MAO-B proteins were each defined by the centroid of the co-crystallized ligands that were present in 2Z5X and 4A79, respectively. We did not remove cofactor FAD during protein preparation and docking. Acacetin and acacetin 7-methyl ether were docked using the Induced Fit docking [37] protocol, applying the standard-precision (SP) docking method. The top 10 poses were kept for analysis. The best docking poses were subjected to binding free-energy calculations using the Prime MM-GBSA module of Schrӧdinger software.

### 4.8. Molecular Dynamics (MD) Simulations

Monoamine oxidase B consists of a two-domain molecular architecture. Each identical monomer consists of 520 amino acids. In this study, we used one monomer (chain A) for the docking and MD simulations. It is reported [38] that the C-terminal helix (residues 498–520) of MAO-B is located in the outer membrane of mitochondria. The MAO-B X-ray crystal structure that we used (PDB ID: 4A79) contains only 501 residues; therefore, we modeled its missing residues that were known to be embedded in lipid bilayer using the Maestro molecular modeling suite [39]. The Protein Preparation Wizard [40] of Maestro was used to prepare the protein structure. We used the Desmond [20] Molecular Dynamics System, ver. 3.6 (Schrödinger) with the OPLS-3 force field and RESPA [41] integrator to perform a 10 ns MD simulation. The best scoring pose of acacetin-7-methyl ether in complex with MAO-B (PDB ID: 4A79) was selected and the C-terminal residues of the protein-ligand complex between amino acids 498 and 520 were embedded into a pre-equilibrated 1-palmitoyl-2-oleoyl-sn-glycero-3-phosphocholine (POPC) membrane and the rest of the system was solvated with TIP3P [42] explicit waters. The whole system was neutralized using sodium chloride (NaCl) and it was set to an ionic strength of 0.15 M. The buffer dimensions of the orthorhombic simulation box were set to 30 × 30 × 70 Å^3^. The solvated system was energy-minimized with the DESMOND minimization algorithm for 2000 iterations when considering a convergence threshold of 1.0 kcal/mol/Å. The constructed system was simulated with the relaxation protocol in Desmond [20]. The protocol involved an initial minimization of the solvent, while keeping restraints on the solute, followed by short MD simulations, including the following steps: (1) Simulation (100 ps) using Brownian dynamics, in the NVT ensemble at 10 K with solute heavy atoms restrained; (2) Simulation (12 ps) in the NVT ensemble using a Berendsen thermostat (10 K) with solute heavy atoms restrained; (3) Simulation (12 ps) in the NPT ensemble using a Berendsen thermostat (10 K) and a Berendsen barostat (1 atm) with non-hydrogen solute atoms restrained; (4) Simulation (12 ps) in the NPT ensemble using a Berendsen thermostat (300 K) and a Berendsen barostat (1 atm) with non-hydrogen solute atoms restrained; and, (5) Simulation (24 ps) in the NPT ensemble using a Berendsen thermostat (300 K) and a Berendsen barostat (1 atm) with no restraints. The production run was carried out using an NPT ensemble at 300 K with Nosé–Hoover temperature coupling [43] and at a constant pressure of 1.01 bar via Martyna–Tobias–Klein pressure coupling [44]. The simulation trajectories (frames) were sampled at intervals of 4.8 ps. We used a timestep of 2.0 fs. The resulting trajectory was analysed using the Simulation Interactions Diagram (SID) utility of Desmond [20]. Finally, the PyMOL 1.4.1 and Maestro 11.5.011 molecular graphics systems were used to visualize the protein-ligand complex and to generate all of the figures.

## 5. Conclusions

Biological screening of constituents of *Turnera diffusa* (damiana) identified three *O*-methyl flavones, namely, acacetin, acacetin 7-methyl ether, and vetulin as selective inhibitors of human MAO-B. Acacetin 7-methyl ether was a highly potent and selective MAO-B inhibitor with >500-fold selectivity towards MAO-B when compared to MAO-A. Further studies suggested that acacetin 7-methyl ether is a reversible competitive inhibitor of MAO-B. Computational docking analysis confirmed the selective binding of acacetin 7-methyl ether to human MAO-B as compared to MAO-A. Acacetin 7-methyl ether may have a potential therapeutic application for the treatment of neurodegenerative disorders, including Parkinson’s disease.

## Figures and Tables

**Figure 1 molecules-24-00810-f001:**
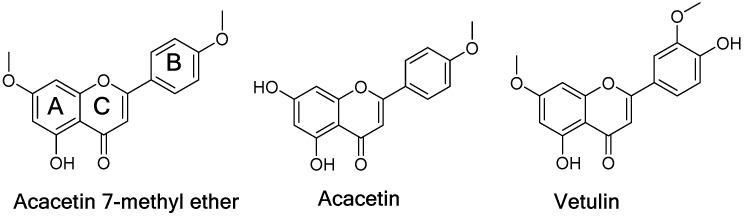
Chemical structure of acacetin, acacetin 7-methyl ether and vetulin.

**Figure 2 molecules-24-00810-f002:**
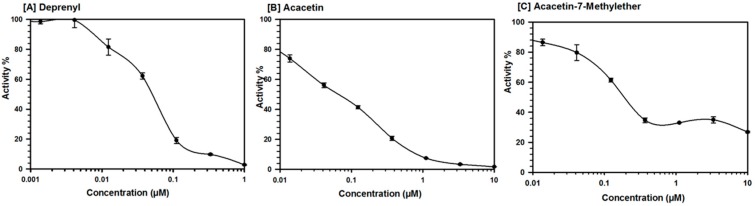
Concentration-dependent inhibition of recombinant human MAO-B by (**A**) deprenyl, (**B**) acacetin, and (**C**) acacetin 7-methyl ether. Each point shows mean ± SD of three observations.

**Figure 3 molecules-24-00810-f003:**
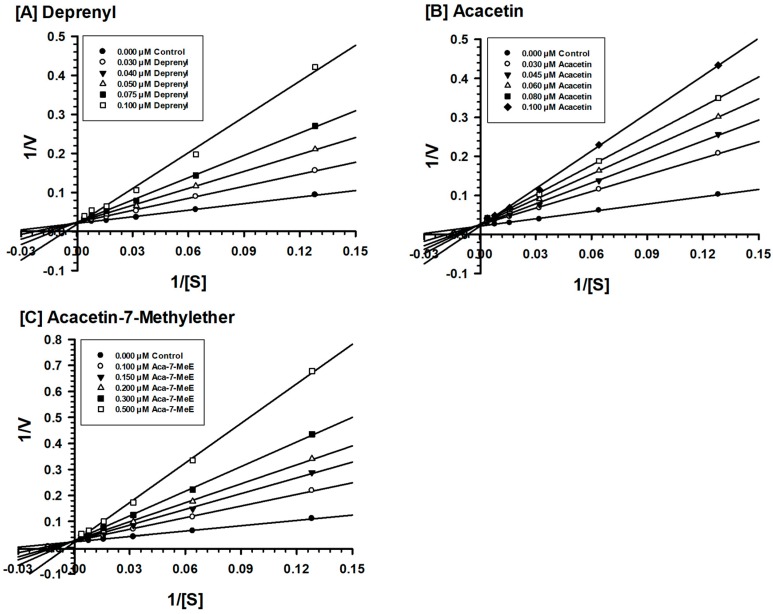
Kinetics characteristics of inhibition of recombinant human MAO-B by (**A**) deprenyl, (**B**) acacetin and (**C**) acacetin 7-methyl ether. Each point shows the mean value of three observations.

**Figure 4 molecules-24-00810-f004:**
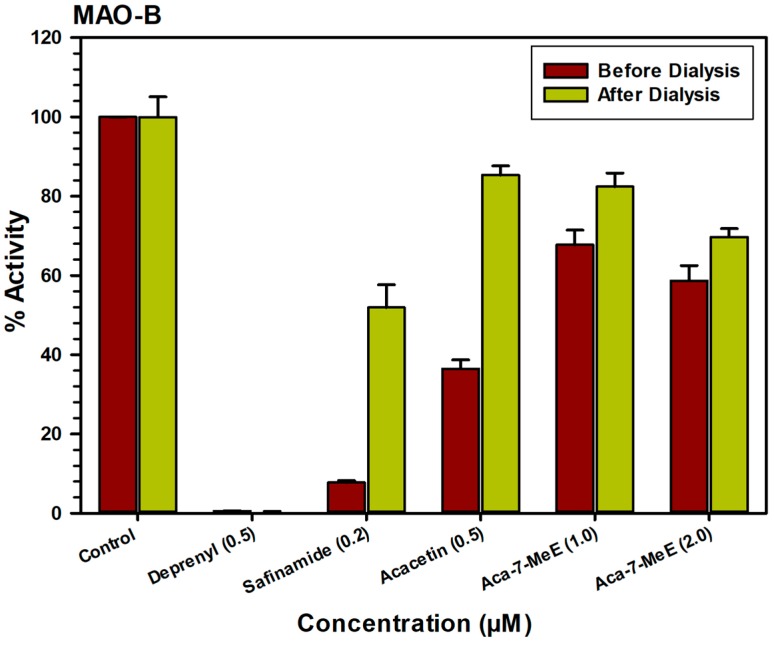
Reversibility assay of recombinant human MAO-B with acacetin (0.5 μM), acacetin 7-methyl ether (Aca 7-MeE) (1.0 and 2.0 μM), deprenyl (0.5 μM), and safinamide (0.2 μM). The remaining activity was expressed as % of activity. Each point shows the mean ± S.D. value of three observations.

**Figure 5 molecules-24-00810-f005:**
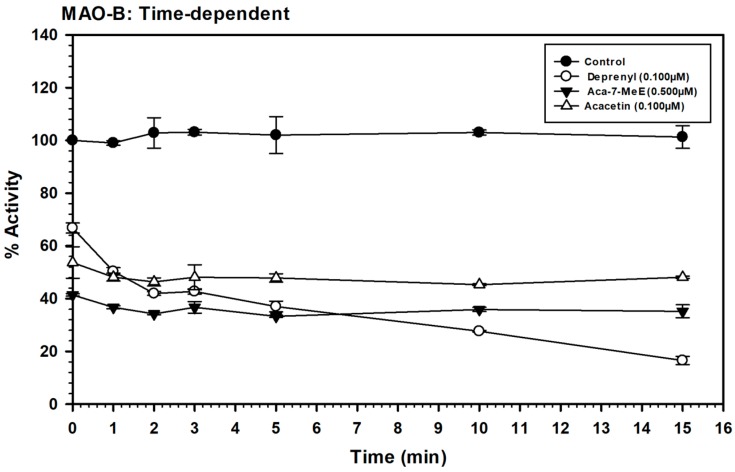
Time-dependent inhibition of recombinant human MAO-B by deprenyl (0.100 μM), acacetin 7-methyl ether (Aca-7-MeE) (0.500 μM), and acacetin (0.100 μM). Each point represents mean ± S.D. of triplicate values.

**Figure 6 molecules-24-00810-f006:**
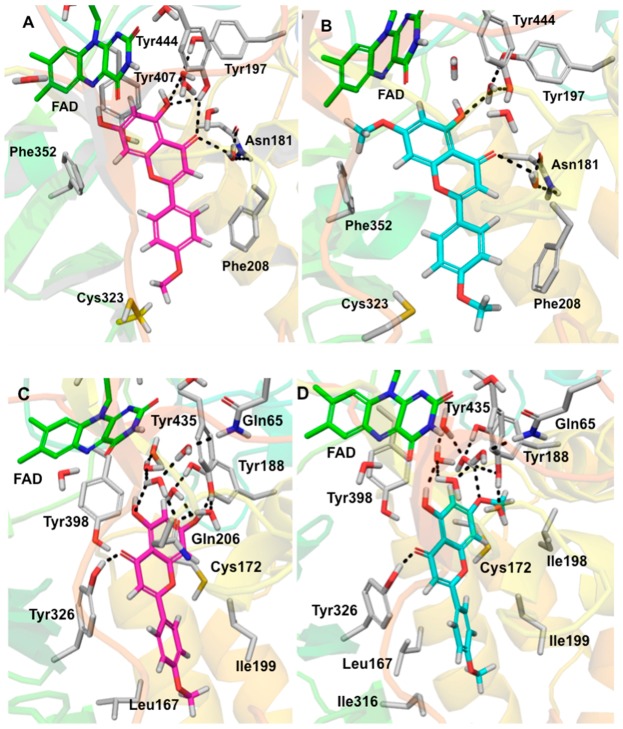
Three-dimensional (3D) protein-ligand interactions of acacetin (C magenta, stick model) and acacetin 7-methyl ether (C cyan, stick model) with the X-ray crystal structures of MAO-A and MAO-B. (**A**) Acacetin with MAO-A, (**B**) acacetin 7-methyl ether with MAO-A, (**C**) acacetin with MAO-B, and (**D**) acacetin 7-methyl ether with MAO-B. FAD (C green, stick model), some crystallographic waters (O red, H gray, stick model), and the important residues of MAO-A and MAO-B (C gray) are also shown. The black dashed lines represent H-bonding.

**Figure 7 molecules-24-00810-f007:**
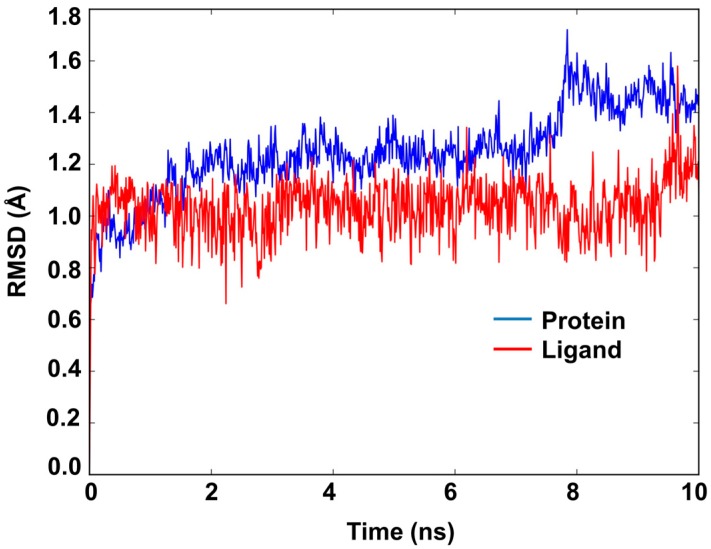
Root Mean Square Deviation (RMSD) plot of atom locations vs. simulation time of MAO-B (protein) and acacetin 7-methyl ether (ligand) for the molecular dynamics (MD) simulation of their interaction complex.

**Figure 8 molecules-24-00810-f008:**
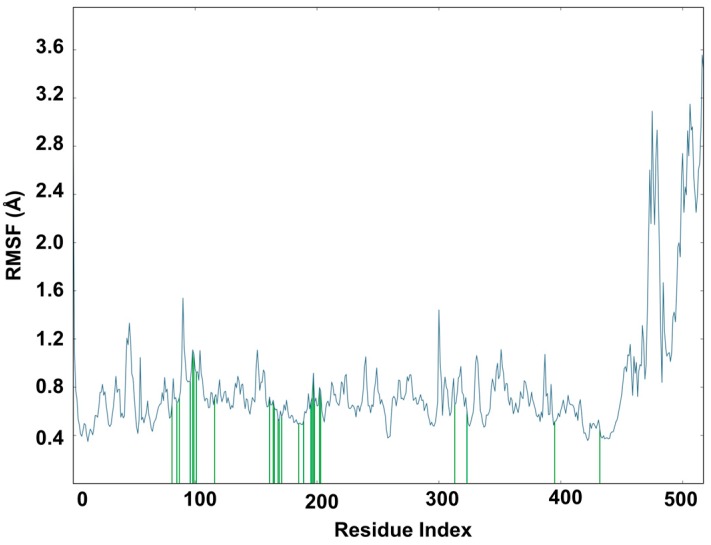
The Root Mean Square Fluctuation (RMSF) plot based on Cα atoms of MAO-B. Protein residues that interact with the acacetin 7-methyl ether is marked with green vertical bars.

**Figure 9 molecules-24-00810-f009:**
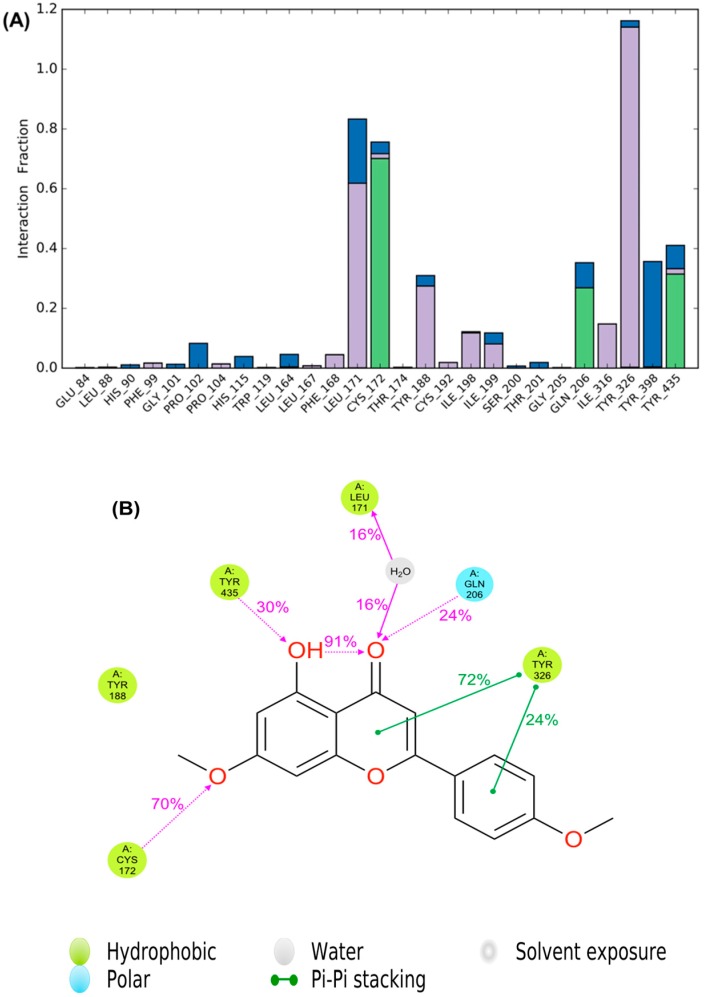
SID (Simulation Interactions Diagram) plots showing the protein-ligand interactions between the amino acid residues of the *h*MAO-B binding site and acacetin 7-methyl ether during the MD simulation. (**A**) The stacked bar charts are categorized as follows: hydrogen bonding (green), hydrophobic interactions (violet), and water bridges (blue) formed. (**B**) A schematic of detailed ligand atom interactions with the protein residues.

**Table 1 molecules-24-00810-t001:** Inhibition of recombinant human Monoamine Amine Oxidase-A and -B by constituents from *T. diffusa*.

Compound	MAO-A IC_50_ (µM) *	MAO-B IC_50_ (µM) *	SI MAO-A/-B
Acacetin 7-methyl ether	>100.00	0.198 ± 0.001	>505.051
Vetulin	18.799 ± 0.291	0.447 ± 0.010	42.056
Apigenin-7-*O*-β-d-(6-*O*-p-coumaroyl) glucoside	22.508 ± 4.440	22.001 ± 1.759	1.023
Echinaticin	21.830 ± 4.367	13.404 ± 0.148	1.703
Tetraphyllin B	-	-	-
Tricin-7-glucoside	50.901 ± 0.506	27.444 ± 0.819	1.854
Diffusavone	19.091 ± 1.450	14.518 ± 0.214	1.314
Turneradiffusin	61.427 ± 3.568	37.250 ± 2.933	1.649
Rhamnosylorientin	34.909 ± 3.887	25.541 ± 2.020	1.366
Rhamnosylvitexin	40.940 ± 6.810	26.286 ± 0.277	1.557
Turneradin	19.169 ± 0.802	13.027 ± 2.142	1.471
Acacetin	0.115 ± 0.004	0.050 ± 0.0025	2.30
Clorgyline	0.0052 ± 0.0001	2.30 ± 0.0570	-
Deprenyl	23.00 ± 1.00	0.051 ± 0.002	450.980
Safinamide ^#^ (ref)	90.00 ± 2.470	0.060 ± 0.005	1500.00

* The IC_50_ values, computed from the dose response inhibition curves, are mean ± S.D. of at least triplicate observations. SI—Selectivity Index-IC_50_ MAO-A/IC_50_ MAO-B inhibition. ^#^ Safinamide data from ref. [18].

**Table 2 molecules-24-00810-t002:** Inhibition/binding affinity constants (Ki) for inhibition of recombinant human MAO-A and MAO-B by acacetin, acacetin 7-methyl ether, and deprenyl.

Compounds	Monoamine Oxidase-B	
	Ki (nM) *	Type of Inhibition
Acacetin 7-methyl ether	45.0 ± 3.0	Competitive/Partially Reversible
Acacetin	36 ± 4.0	Competitive/Reversible
Deprenyl	29 ± 6.3	Mixed/Irreversible

* Values are mean ± S.D. of triplicate experiments.

**Table 3 molecules-24-00810-t003:** Glide docking scores and binding free energies of acacetin and acacetin 7-methyl ether to MAO-A and MAO-B.

Compound	IC_50_ (µM) *		Glide Docking Score (kcal/mol)		Binding Free-Energies (kcal/mol)	
	MAO-A	MAO-B	MAO-A	MAO-B	MAO-A	MAO-B
Acacetin	0.115	0.049	−10.685	−11.890	−56.303	−67.205
Acacetin 7-methyl ether	>100	0.198	−9.085	−10.708	−31.791	−67.494

* Values are mean ± S.D. of triplicate experiments.

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
