# Peer review of "Selective Inhibition of Human Monoamine Oxidase B by Acacetin 7-Methyl Ether Isolated from *Turnera diffusa* (Damiana)"

_molecules, 2019, doi:10.3390/molecules24040810_

Reviewer 1 Report

Constituents isolated from Turnera diffusa were evaluated as inhibitors of recombinant hMAO-A and-B . Among them, acacetin 7-methyl ether was identified as a potent and highly selective inhibitor of hMAO-B. Selective binding and mechanism of inhibition of acacetin 7-methyl were investigated through computational and kinetic studies

The paper lacks originality since acacetin, and many other flavonoids, have been already described as reversible MAO inhibitors. The present study addressed chiefly the main biochemical and binding properties of the 7-methyl ether of acacetin which resulted a less potent but more selective MAO B inhibitor than its parent compound. Research methodologies, biological and computational results overlay with the ones reported recently by Tekwani and coll. on acacetiin. As a consequence, no significantly new insights into binding, kinetic, reversibility and selectivity of MAO inhibition by flavonoids emerged from the present study.

However, the paper is well written, presented and discussed and the results, even not so original, worth publishing. The paper can be accepted for publication upon suitable modifications coming from the observations, remarks and suggestions reported below.

1. Why was X-ray 3D structure of the pioglitazone-hMAO-complex (PDB, 4A79) chosen for docking studies? Many other complexes with inhibitors structurally closer to acacetin, are indeed available in PDB ( see Binda C et al J. Med. Chem., 2007, 50 , pp 5848–5852)
2. The results obtained in the computational study are surely interesting, but some interpretations should be presented in a more conservative way. Further, more informative insights into the binding mode of acacetin 7-methyl ether and into the stability over time of its main interactions with hMAO-B could have been acquired by means of a molecular dynamic study
3. In the title, and other parts of the MS as well, neurodegenerative disorders and Parkinson’s disease (PD) seem to be considered as distinct pathologies. PD is indeed a neurodegenerative disease as indirectly also quoted/recognized in the discussion : “ … in the treatment of neurodegenerative disorders, including Parkinson’s disease [20] “
4. In the references publication years are often not in bold.
5. p.1, l.42 : what does” inactivation” mean? Better to use deaminative oxidation
6. p.2, l. 47: Differently from PD, In AD MAO inhibitors are not used as drugs.
7. In the legenda of Table 1, SI should be more precisely defined as the ratio between MAO A and MAO B IC50s
8. In the discussion,p.8, l.229. MAO B inhibitors are not used alone in PD, at least in gold standard therapies
9. p.8, l. 235: safinamide and not sulfinamide

Author Response

Reviewer 1 

 Constituents isolated from Turnera diffusa were evaluated as inhibitors of recombinant hMAO-A and-B. Among them, acacetin 7-methyl ether was identified as a potent and highly selective inhibitor of hMAO-B. Selective binding and mechanism of inhibition of acacetin 7-methyl were investigated through computational and kinetic studies 

The paper lacks originality since acacetin, and many other flavonoids, have been already described as reversible MAO inhibitors. The present study addressed chiefly the main biochemical and binding properties of the 7-methyl ether of acacetin which resulted a less potent but more selective MAO B inhibitor than its parent compound. Research methodologies, biological and computational results overlay with the ones reported recently by Tekwani and coll. on acacetiin. As a consequence, no significantly new insights into binding, kinetic, reversibility and selectivity of MAO inhibition by flavonoids emerged from the present study. 

However, the paper is well written, presented and discussed and the results, even not so original, worth publishing. The paper can be accepted for publication upon suitable modifications coming from the observations, remarks and suggestions reported below. 

1. Why was X-ray 3D structure of the pioglitazone-hMAO-complex (PDB, 4A79) chosen for docking studies? Many other complexes with inhibitors structurally closer to acacetin, are indeed available in PDB (see Binda C et al J. Med. Chem., 2007, 50 , pp 5848–5852) 

Response: Two X-ray crystal structures of hMAO-B co-crystallized with coumarin derivatives are available in the Protein Data Bank (accession codes 2V60 and 2V61, Binda et al. J. Med. Chem. 2007, 50, 23, 5848-5852). However, to perform the docking and MD simulations we selected the X-ray crystal structures of hMAO in complex with pioglitazone (PDB ID: 4A79) because of the following reasons:

1.    We previously published docking study of acacetin for MAO-B using the protein structure PDB ID: 4A79 (J. Nat. Prod. 2016, 79, 10, 2538-2544); we also used the same X-ray crystal structure for consistency in our reports.

2.    We were also interested in comparing our docking results with Lee, H.W. 2017 (Int J Biol Macromol. 2017;104: 547-553). They also used PDB ID:4A79 for acacetin and its analogs for docking. 

In addition, we have calculated the RMSD between the 4A79 and 2V61 X-ray crystal structures of the hMAO-B protein and found only 0.17 Å RMSD between the overall protein structures. The co-crystalized ligand of 2V61 also overlaps with the co-crystalized ligand (pioglitazone) of 4A79. Therefore, we believe we made an acceptable choice of PDB ID: 4A79.

2. The results obtained in the computational study are surely interesting, but some interpretations should be presented in a more conservative way. Further, more informative insights into the binding mode of acacetin 7-methyl ether and into the stability over time of its main interactions with hMAO-B could have been acquired by means of a molecular dynamic study 

Response: We agree with the reviewer’s suggestion. Docking alone may not shed full insight into the binding mode and dynamics of acacetin 7-methyl ether within hMAO-B. Therefore we have further carried out a 10 ns MD simulation and incorporated our results and insights into the revised manuscript. We made sure to be conservative in our interpretation of the results.

3. In the title, and other parts of the MS as well, neurodegenerative disorders and Parkinson’s disease (PD) seem to be considered as distinct pathologies. PD is indeed a neurodegenerative disease as indirectly also quoted/recognized in the discussion: “…in the treatment of neurodegenerative disorders, including Parkinson’s disease [20] “ 

Response: Thanks for the suggestion. We made adequate corrections in the revised manuscript. We changed “neurodegenerative disorders and Parkinson’s disease” to “neurodegenerative disorders, including Parkinson’s disease” in the entire manuscript.

4. In the references publication years are often not in bold. 

Response: References, publication years are corrected according to molecules guidelines.

5. p.1, l.42: what does” inactivation” mean? Better to use deaminative oxidation 

Response: In Page 1. Line 42 inactivation word replaced with oxidative deamination.

6. p.2, l. 47: Differently from PD, In AD MAO inhibitors are not used as drugs. 

Response: The sentence revised as “MAO-B inhibitors are useful for treatment Parkinson's disease and also in combination for treatment of Alzheimer’s Disease”

7. In the legenda of Table 1, SI should be more precisely defined as the ratio between MAO A and MAO B IC50s 

Response: revised as - SI- Selectivity Index-IC50MAO-A/ IC50MAO-B inhibition

8. In the discussion, p.8, l.229. MAO B inhibitors are not used alone in PD, at least in gold standard therapies 

Response: sentence revised as “MAO-B inhibitors with potential therapeutic application for treatment of neurodegenerative disorders including Parkinson’s disease in combination with standard drugs

9. p.8, l. 235: safinamide and not sulfinamide 

Response: Page 8 line 235 safinamide is corrected.

Reviewer 2 Report

This manuscript is describing about selective inhibition against hMAO-B by acacetin acacetin 7-methyl ether (AME) isolated from Turnera diffusa (damiana). The findings are valuable that AME was potent and selective for MAO-B with SI >500 and competitive reversible. However, this manuscript should be reconsidered after major revision.

1) Acacetin and Acacetin 7-O-(6-O-malonylglucoside) were reported in other papers [23, 25]. Therefore, those results should be mentioned in the view of differences and originality of this study should be included in 'Introduction'. At the same time, those should be compared and discussed in detail in 'Discussion' 

2) This study tested irreversible inhibitors. However, reversible reference compounds for MAO-A and MAO-B should be included, because AME is reversible.

3) There is no experiments about clinical or in vivo in this manuscript. Therefore, title should be shortened by deleting 'Relevant to Neurodegenerative Disorders and Parkinson’s Disease'.

4) In Fig. 2 (C), perturbation of concentration-dependent inhibition should be overcome at 1~10 uM.

5) In Fig. 3 (A) and Table 2, deprenyl looks 'competitive' rather than 'mixed'.

6) In Fig. 4, Concentration for deprenyl and acacetin was 0.5 uM, which is about 10 x IC50. Activities for 'Before dialysis' were ~0 and ~40%, respectively. Concentrations for AME were also 5 x IC50 and 10 x IC50, however, activity for 'Before dialysis' were ~60% and ~70%, respectively.  Why did you get big different data? Theoretically, data for 'Before dialysis' at the 10 x IC50 should be much lower than those you got.   

 - In Fig. 4, why did you use the high concentration of deprenyl and acacetin?

- In Fig. 4, reversible reference should be included.

 - Acacetine -> acacetin

7) In Fig. 5, deprenyl was still decreasing at 15 min. Preincubation time should be longer to 30 min. If so, IC50 and Ki should be smaller than the values you got.

8) Lines 19, 22; IC50- -> IC50 =

    Line 77, ref citation style was incorrect.

    Line 78, IC50 = > 100 uM -> IC50 > 100 uM

    Lines 131, 137, 138, 307; 0.100, 0.500 -> 0.10, 0.50

    Line 299; Mm -> mM

Author Response

Review 2

Comments and Suggestions for Authors

This manuscript is describing about selective inhibition against hMAO-B by acacetin acacetin 7-methyl ether (AME) isolated from Turnera diffusa (damiana). The findings are valuable that AME was potent and selective for MAO-B with SI >500 and competitive reversible. However, this manuscript should be reconsidered after major revision.

1) Acacetin and Acacetin 7-O-(6-O-malonylglucoside) were reported in other papers [23, 25]. Therefore, those results should be mentioned in the view of differences and originality of this study should be included in 'Introduction'. At the same time, those should be compared and discussed in detail in 'Discussion' 

Response: We have added the recommended references about Acacetin and Acacetin 7-O-(6-O-malonylglucoside) as references [11, 12] in the revised manuscript and provided the details information in introduction and Discussion section by comparing with our findings.

2) This study tested irreversible inhibitors. However, reversible reference compounds for MAO-A and MAO-B should be included, because AME is reversible.

Response:In this study, we have studied selective MAO-B potent inhibition by acac-7-methyl ether and we have included reversible reference compound (safinamide known MAO-B reversible inhibitor) [28]. 

3) There is no experiments about clinical or in vivo in this manuscript. Therefore, title should be shortened by deleting 'Relevant to Neurodegenerative Disorders and Parkinson’s Disease'.

Response:The title has been revised as suggested.

4) In Fig. 2 (C), perturbation of concentration-dependent inhibition should be overcome at 1~10 uM.

Response:In fig 2 (C), concentration –dependent graph showed approx. 80% inhibition and 20% enzyme activity remains. 

5) In Fig. 3 (A) and Table 2, deprenyl looks 'competitive' rather than 'mixed'.

Response: Deprenyl is MAO-B selective potent irreversible mixed type inhibitor. Irreversible inhibitors are not inhibiting enzyme activity in a competitive way. New references included related to deprenyl irreversible inhibition [26, 27].   

6) In Fig. 4, Concentration for deprenyl and acacetin was 0.5 uM, which is about 10 x IC50. Activities for 'Before dialysis' were ~0 and ~40%, respectively. Concentrations for AME were also 5 x IC50 and 10 x IC50, however, activity for 'Before dialysis' were ~60% and ~70%, respectively.  Why did you get big different data? Theoretically, data for 'Before dialysis' at the 10 x IC50 should be much lower than those you got.   

 - In Fig. 4, why did you use the high concentration of deprenyl and acacetin?

- In Fig. 4, reversible reference should be included.

 - Acacetine -> acacetin

Response:We used higher concentration of deprenyl and acacetin in reversibility assay for maximum binding; 

- Because of enzyme concentration is high in reaction mixture to facilitate formation of enzyme + inhibitor (E+I) complex.

- Reversible reference compound safinamide included in fig 4 reversibility graph.

- Changed acacetin typo error in X axis of reversibility fig 4.

7) In Fig. 5, deprenyl was still decreasing at 15 min. Preincubation time should be longer to 30 min. If so, IC50 and Ki should be smaller than the values you got.

Response:In time dependent experiment, we have investigated that compounds are inhibiting enzyme activity time dependently or not? And in fig 5, within 5-minute results showed inhibitors are time depended or not.

8) Lines 19, 22; IC50- -> IC50 =

    Line 77, ref citation style was incorrect.

    Line 78, IC50 = > 100 uM -> IC50 > 100 uM

    Lines 131, 137, 138, 307; 0.100, 0.500 -> 0.10, 0.50

    Line 299; Mm -> mM

Response:-In line 19, 22 (–) replaced with (=).

- Line 77, reference style changed.

- Line 78 – changed to =.

- Lines 131, 137, 138, 307; 0.100-0.500 - 0.10-0.50.

- Line 299 Mm changed to mM.

Reviewer 3 Report

molecules-427277: Selective Inhibition of Human Monoamine Oxidase B by Acacetin 7-Methyl Ether and Molecular Docking Studies Relevant to Neurodegenerative Disorders and Parkinson’s Disease This paper performed the kinetic studies of the promising monoamine oxidase B (MAO-B) inhibitor from damiana, a plant used in alternative medicine for treatment of various diseases. The information of this paper will help the development of a new class of MAO-B inhibitor, which has been reevaluated in the treatment of Parkinson's disease. The experiments were performed very carefully and almost all the necessary kinetic data of the inhibitor were presented. Hence, this manuscript is fairly ready for the publication. Below are some suggestions, which may help to improve the value of this original paper. 1. Discussion on the potency and efficacy. From Table 1, it is true that acacetin 7-methyl ether has high potency (low IC50) and selectivity (high SI). However, from Fig. 2C, the maximum inhibition by is around 20%. The inhibition of MAO-B could not reach to 0% in a high acacetin 7-methyl ether concentration of 10 microM. In case of deprenly (clinically used), almost 0 % was achieved in 1 microM, and in case of acacetin, it was achieved around 10 microM. This means efficacy of acacetin 7-methyl ether is rather low. This point should be discussed. 2. The title including clinical application may be overestimating. Although the title includes "Relevant to Neurodegenerative Disorders and Parkinson’s Disease", no data of clinical application or in vivo animal experiments were presented. For example, pharmacokinetics (ie. absorption, tissue distribution, and so on) of acacetin 7-methyl in vivo is necessary for the clinical trial. Hence, a simple and informative title such as " Selective Inhibition of Human Monoamine Oxidase B by Acacetin 7-Methyl Ether from Turnera diffusa (damiana)" may be more suitable. a. Related to above "Major Points 2", the metabolism of acacetin 7-methyl ether in vivo may be briefly discussed. Is there a possibility that acacetin 7-methyl ether may be rapidly metabolized to acacetin in vivo? b. The purity should be expressed in %. In line 269, the purity of the chemicals should be shown as "% purity". End of File.

Author Response

Review 3

Comments and Suggestions for Authors; molecules-427277: Selective Inhibition of Human Monoamine Oxidase B by Acacetin 7-Methyl Ether and Molecular Docking Studies Relevant to Neurodegenerative Disorders and Parkinson’s Disease This paper performed the kinetic studies of the promising monoamine oxidase B (MAO-B) inhibitor from damiana, a plant used in alternative medicine for treatment of various diseases. The information of this paper will help the development of a new class of MAO-B inhibitor, which has been reevaluated in the treatment of Parkinson's disease. The experiments were performed very carefully and almost all the necessary kinetic data of the inhibitor were presented. Hence, this manuscript is fairly ready for the publication. Below are some suggestions, which may help to improve the value of this original paper. 

1. Discussion on the potency and efficacy. From Table 1, it is true that acacetin 7-methyl ether has high potency (low IC50) and selectivity (high SI). However, from Fig. 2C, the maximum inhibition by is around 20%. The inhibition of MAO-B could not reach to 0% in a high acacetin 7-methyl ether concentration of 10 microM. In case of deprenly (clinically used), almost 0 % was achieved in 1 microM, and in case of acacetin, it was achieved around 10 microM. This means efficacy of acacetin 7-methyl ether is rather low. This point should be discussed. 

Response:In Figure 2C showed 80% inhibition and remains 20% enzyme activity to confirm inhibition effect of acaetin-7 methyl ether in table 1.

2. The title including clinical application may be overestimating. Although the title includes "Relevant to Neurodegenerative Disorders and Parkinson’s Disease", no data of clinical application or in vivo animal experiments were presented. For example, pharmacokinetics (ie. absorption, tissue distribution, and so on) of acacetin 7-methyl in vivo is necessary for the clinical trial. Hence, a simple and informative title such as " Selective Inhibition of Human Monoamine Oxidase B by Acacetin 7-Methyl Ether from Turnera diffusa (damiana)"may be more suitable. 

Response:We have modified title of this manuscript.                                                                                

Related to above "Major Points 2", the metabolism of acacetin 7-methyl ether in vivo may be briefly discussed. Is there a possibility that acacetin 7-methyl ether may be rapidly metabolized to acacetin in vivo? 

Response:Very limited studies have been done with metabolism of Acacetin in vivo. This will be a plan for our future studies.  

b. The purity should be expressed in %. In line 269, the purity of the chemicals should be shown as "% purity". 

Response:The purity of all compounds are expressed in %. 

Round  2

Reviewer 2 Report

The manusript was revised partly according to the comments. However, it has still serious flaws and additional experimets should be needed. It is highly regrettable that some of the responses to the comments were not addressed well. Main points are:

1) In Fig 2., incomplete inhibition was not cleared at 1, 5, and 10 uM of AME. It should be checked why the inhibition was not decreased to low activity. The curve was not normal at the range. The value 20% is not successful to determine the inhibition. I am not sure, however, it may be related to the low inhibition degree by AME before dialysis in Fig. 4.

2) In Fig 3, the plot of yours looks competitve. Anyway, the plot of the irreversible inhibitor should be replaced by one of a reversible inhibitor such as safinamide.

3) In Fig. 4, the inhibition degree by AME before dialysis was too high (~70% activity), comparing to other refererence. So, the result was not sufficient to show the reversibility.

Author Response

Reviewer 2

Comments and Suggestions for   Authors

The manuscript was revised partly according to the comments. However, it has still serious flaws and additional experiments should be needed. It is highly regrettable that some of the responses to the comments were not addressed well. Main points are:

1) In Fig 2., incomplete inhibition was not cleared at 1, 5, and 10 uM of AME. It should be checked why the inhibition was not decreased to low activity. The curve was not normal at the range. The value 20% is not successful to determine the inhibition. I am not sure, however, it may be related to the low inhibition degree by AME before dialysis in Fig. 4.

The concertation-dependent inhibition profile showed a plateau at ~80% inhibition (20% remaining activity) potential due to solubility issues with the compound at higher concentration. This has been mentioned in the revised manuscript    

2) In Fig 3, the plot of yours looks competitive. Anyway, the plot of the irreversible inhibitor should be replaced by one of a reversible inhibitor such as safinamide.

The enzyme kinetics results with acacetin, which also shows the competitive inhibition of MAO-B, have been included for comparison. It may not be necessary to include enzyme kinetics data with another reversible inhibitor    

3) In Fig. 4, the inhibition degree by AME before dialysis was too high (~70% activity), comparing to other reference. So, the result was not sufficient to show the reversibility.

For equilibrium dialysis assay high concentration of the enzyme protein is used and due to potential solubility issues with acacetin 7-methyl ether at higher concentrations about 30 and 40 % inhibition were noticed at 1 and 2 mM concentrations respectively. Recovery after dialysis was significant to conclude that binding of acacetin 7-methyl ether wit MAO-B is reversible.